

# Modelling the effects of perceived system quality and personal innovativeness on the intention to use metaverse: a structural equation modelling approach

Sultan Hammad Alshammari and Mohammed Habib Alshammari

Department of Educational Technology, College of Education, University of Ha'il, Ha'il, Saudi Arabia

## ABSTRACT

The metaverse, an interactive and immersive 3D virtual environment, has recently become popular and is widely used in several fields, including education. However, the successful use of metaverse relies on the extent to which users intend to adopt and use it. Close examination of this critical issue reveals a lack of research that examines the effects of certain factors on users' intentions toward using metaverses. Thus, this study extends the technology acceptance model by integrating two constructs— perceived system quality and students' personal innovativeness. Using a survey to collect data, 164 responses were received from students at the University of Ha'il in Saudi Arabia. Two steps in structural equation modelling (SEM) using the AMOS software were applied to analyse the data and test the research hypotheses. The results revealed that perceived system quality had a significant effect on students' intentions to use metaverses through perceived ease of use. Furthermore, personal innovativeness had a significant effect on students' intentions through the perceived usefulness of the metaverse. In addition, perceived usefulness affected students' intentions to use a metaverse. Surprisingly, perceived ease of use had an insignificant effect on students' intentions to use the metaverse. Although the proposed model and its findings contribute to the technology acceptance model (TAM) literature, the study's practical value is significant because it can help educational policymakers and authorities to understand the effect of each factor and plan future strategies. Additionally, the findings of this study can assist practitioners, designers, and developers in designing and promoting the utilisation of metaverses.

## INTRODUCTION

The metaverse is viewed as the coming Internet generation that can rapidly change users' communication and interaction with others worldwide (*Hwang & Chien, 2022*). An interactive and immersive virtual environment using 3D that assists in the digital simulation of the physical environment, the metaverse has generated significant interest (*Ning et al., 2023*; *Alkhwaldi, 2024*). The metaverse was developed using several emerging technologies, such as augmented reality, artificial intelligence, digital twins, virtual reality, blockchain, 5G, and extended and mixed reality (*Wang et al., 2022*). It has advanced

Corresponding author
Mohammed Habib Alshammari,
m.habib@uoh.edu.sa

features in its decentralised, interactive nature and its effectiveness in representing a virtual community (*Hwang & Chien, 2022*). It enables a wide ranges of interactive environments where users can access digital avatars that enable them to communicate with others (*Cheng et al., 2022*). Persistent and perpetual spaces enable users to communicate and engage in several different activities for instance, work, tourism, conferences and education (*Skalidis, Muller & Fournier, 2022*). Furthermore, the metaverse is designed and built based on decentralised technologies that enable users to possess autonomy and ownership (*Ng, 2022*).

Moreover, the utilisation of the metaverse in education can enhance virtual online learning experiences and make it easier for instructors to provide virtual classes which imitate traditional face-to-face classes (*Teng et al., 2022*). In addition, the metaverse could have a higher level of customisation and creativity and less associated risk that could attract students' interaction and motivation and boost their engagement, thereby expanding the activities of traditional learning methods by providing experiences and opportunities that may not be feasible using other technologies (*Ng, 2022*). Therefore, compared to prior technologies, using metaverse in educational field might allow better interaction in the learning environment, recreating cognitive and emotional processes and closely mimicking the experiences of face-to-face classes. Furthermore, metaverse learning contributes significantly to enhancing learners' motivation and immersion (*Akour et al., 2022*). For example, it provides students with the opportunity to attend simulated and virtual classrooms and interact with their instructors and classmates using personalised digital avatars.

The development of metaverse technologies will create up to 23.3 million new jobs worldwide by the year of 2020, and the revenue growth of education will reach about 294.2 billion dollars (*Ababkova, Pokrovskaia & Haj Bara, 2023*). Furthermore, the estimation of the metaverse in the educational market will grow at a CAGR "Compound Annual Growth Rate" of 42.9% in the year of 2021 to over 13$ billion by the year 2026. Additionally, A Holon IQ report declared that the global spending on metaverse in education will increase from 1.8$ billion in year 2021 to 12.6$ billion by year 2025 (*Anderson, 2023*). The effectiveness, sustainability, and value of metaverse might hinder user acceptance (*Wang & Shin, 2022*). A growing community of users has been accepting the benefits of metaverses (*Ni & Cheung, 2023*). In addition, accepting a particular technology among users is an essential factor in its development, implementation, and investment. In a metaverse context, practitioners and designers should understand the factors driving user acceptance prior to the investment, development, and implementation of this emerging technology. However, most current studies on the metaverse have focused on technical issues and aspects (*Du et al., 2023*; *Lee & Gu, 2022*). Few published studies had focused on factors that may affect users' acceptance (*Al-Adwan et al., 2023*; *İbili et al., 2023*; *Wu & Yu, 2023*). Thus, there is a need to comprehend the factors which determine the metaverses acceptance among their users, a critical issue during this period (*Alkhwaldi, 2023*; *Maghaydah et al., 2024*). Various variables, such as social and psychological constructs, social norms and self-efficacy (*İbili et al., 2023*), trust (*Jeong & Kim, 2023*), and social

media experiences (*Pan, Jung & Suo, 2023*), might determine metaverse acceptance among its users (*Wu & Yu, 2023*). Nevertheless, little is known about the effect of personal innovativeness and system quality on the user acceptance of metaverses. System quality is defined as the characteristics of an information system in terms of its easiness for usage, system reliability and flexibility, response time, sophistication and intuitiveness (*Petter, DeLone & McLean, 2008*). It is also related to the measure of information systems in terms of its design and technical prospectives (*Gable, Sedera & Chan, 2008*). According to IS success model, system quality is an essential success factor which could influence users' intention to use (*DeLone & McLean, 2003*). With the context of metaverse, it is defined as the characteristics of metaverse in terms of its ease of use, flexibility and reliability, response time, technical and design features. Despite the significant number of studies on system quality, very few studies have assessed its effective role in students' acceptance of metaverse. On the other hand, personal innovativeness stands for a user's willingness to experience a new established technology (*Agarwal & Prasad, 1998*). In the context of metaverse, personal innovativeness refers to the students' willing to experience metaverse. Even though personal innovativeness has been assessed with many technologies such as with massive open online courses (MOOCS) (*Gupta, 2021*), learning management system (LMS) (*Abdul Rahman, Umar & Abdullah, 2020*), adoption of online courses (*Amid & Din, 2021*) *etc.*, the effect of personal innovativeness on metaverse has received little attention and not yet understood. Furthermore, *İbili et al. (2023)* call for further study to extend a technology acceptance model and examine the effect of perceived system quality and personal innovativeness on students' acceptance of metaverse. Additionally, it is unclear whether these factors, namely perceived system quality and personal innovativeness, could affect students' intention toward accepting the use of a metaverse. Moreover, thus far, no study has assessed the effects of metaverse system quality and students' personal innovativeness on users' intentions to use metaverses. To address this research gap, the current study was conducted to determine whether these factors, students' personal innovativeness, and metaverse system quality affect the acceptance of metaverses.

The current study contributes to existing research that focuses on various aspects of metaverse acceptance. First, it confirms the empirical application and validity of the technology assistance model's (TAM's) original constructs in a metaverse context. Second, the study novelty lies in its proposed conceptual model incorporating the constructs of personal innovativeness and perceived system quality into TAM constructs. Thus, this study makes valuable contribution. Third, it might be a valuable study for practitioners and designers who wish to comprehend the factors which influence the user's acceptance of a metaverse and to work towards optimising the metaverse, which could lead to an increase in the user's acceptance of it. In the following sections, this study focused on discussing these aspects; the literature review related to studies conducted on metaverse, the research hypothesis and proposed model, procedures of methodology in regard to data collection, measures and data analysis, presenting the findings, discussing and comparing them with previous studies.

 

## LITERATURE REVIEW

Several studies have attempted to reach a clear and unique definition of the metaverse (*Narin, 2021*). *Lee & Kim (2022)* defined a metaverse as an immersive, permanently minimised reality environment where users and objects could collaborate, communicate, and interact with no restrictions on space and time by using a digital avatar. It seeks to produce a mirror of the real world and integrate both real and virtual worlds into all spaces of life. Further, several emerging technologies have played significant roles in accelerating the growth of metaverses such as: (1) Augmented and virtual reality technologies that provide 3D immersive experiences with multi-sensor communication; (2) Enhancing mobile broadband, 5G/6G, and ultra-reliable low-latency communications to provide a higher structure for communicating in systems; (3) The blockchain and artificial intelligence (AI) that enable the creation of large contents and the use of non-fungible tokens (NFTs) for supporting digital resources (*Lee & Kim, 2022*). All these technologies have made significant advances in the metaverse industry. Furthermore, *Wang et al. (2023)* highlighted the power services of the metaverse which include modification of video games and earning digital assets and money, immersive social experiences, online collaboration and learning, 3D design and simulation of physical objects, creation of content, and commerce using the metaverse through the support of NFTs and blockchain (*e.g.*, trade of arts, digital music, content, and applying virtual works and business).

The metaverse usage for educational purposes in Arab counties has been increased recently. A study of *Salloum et al. (2023)* proposed a model for assessing the continuous intention to use metaverse in higher education in Oman, and the findings showed that innovative academic environment could affect students' and teachers' attitude toward new technology. Understanding how the function of metaverse such as innovative education tools could affect students' views regarding using it, assisting institutions in higher education to set and develop new regulations which could enhance their learning process.

Additionally, several challenges are associated with metaverse use (*Chen, 2022*). Metaverse behaviours can increase mental and physical health risks (*Tlili, Huang & Kinshuk, 2023*), stress, loneliness and cognitive load (*Bibri & Allam, 2022*; *Oh et al., 2023*), and virtual world addiction (*Bojic, 2022*). Furthermore, the private stored information of users in a metaverse can be hacked once they use it (*Tlili, Huang & Kinshuk, 2023*). Additionally, metaverse use can lead to some ethical concerns, for instance, inequalities, bullying, and discrimination (*Dwivedi et al., 2022*).

Although academics in the field of education have not yet adequately considered metaverse use, some researchers have begun conducting studies on this emerging technology and have highlighted several findings (*Guo & Gao, 2022*). Studies that have assessed the educational utilisation of metaverses have mostly focused on language learning, medical education, science education, physical education, and VR/AR technologies. For instance, *Suh & Ahn*'s *(2022)* study assessed the experiences of students in elementary schools with metaverse to evaluate if it is suitable for learner-centred constructivist pedagogy. *Ortega-Rodríguez (2022)* conducted a study to evaluate the benefits and pedagogical challenges of education using a metaverse. Moreover,

*Jovanović & Milosavljević (2022)* assessed both the disadvantages and advantages of learning in metaverse and the setting of traditional classes. *Erturk & Reynolds (2020)* investigated the effects of immersive media in metaverse environments and how it could be a convenient tool that supports learning. Moreover, the metaverse adoption practices by instructors and students has gained further attention. For instance, *Almarzouqi, Aburayya & Salloum (2022)* conducted a study that assessed postgraduate and undergraduate students' perceptions of metaverse adoption for use in the medical field. Similarly, *Lee & Hwang (2022)* used the metaverse to link the experience of English teachers to assess their readiness and the challenges associated with using it. *Mustafa (2022)* assessed university instructors' and students' opinions regarding the metaverse and its support system during its use. *Teng et al. (2022)* investigated the factors that influence students' intentions toward the metaverse use of basketball.

For the metaverse to be successfully implemented, it is essential to assess the factors that may contribute to its adoption and acceptance (*Al-Qaysi, Mohamad-Nordin & Al-Emran, 2020*). Thus, an urgent need exists to understand the factors which contribute to metaverse uses among users. Developed by *Davis (1989),* the TAM is a robust and valid theoretical framework for assessing the contributing factors which may affect the metaverse adoption (*Ren et al., 2022*). It can be considered the most utilised model in information systems to examine the core factors that may affect users' adoption of a technology, either after or prior to technology use (*Mascret et al., 2022*). Beside, TAM has proved its effectiveness when compared with other theoretical models (*Al-Qaysi, Mohamad-Nordin & Al-Emran, 2020*). *Davis (1989)* developed a model containing four main constructs as determinants: perceived usefulness (PU), perceived ease of use (PEU), attitude (ATT), and behavior intention (BI). PU can be defined as the number of users who believe that using a metaverse is worthy of productivity and effectiveness. PEU can be defined as the users' level of belief that the use of a metaverse may require less time and effort. ATT refers to users' level of belief that using a metaverse would interest them. BI indicates that the user intends to continually use the metaverse. In the TAM, PU and PEU are considered the main determinants of ATT, and ATT is the determinant of users' intention toward use. However, ATT was eliminated from the current proposed model as most TAM models proposed direct effect of PU and PEU on BI, without attitude (*Venkatesh et al., 2003*; *van der Heijden, 2004*; *Sánchez-Prieto, Olmos-Migueláñez & García-Peñalvo, 2017*). Moreover, the TAM can be extended to integrate external factors that could affect its determinants' factors indirectly or directly.

In the TAM and metaverse context, some studies have attempted to extend the TAM to comprehend the factors affecting users' intentions to accept and use a metaverse. *Wang & Shin (2022)* extended TAM to determine the effects of certain factors on accepting and using metaverse; for instance, they included situational teaching, personalised learning, technical maturity, and perceived risk. *Afkar et al. (2022)* assessed users' intention towards using a metaverse by extending TAM with additional external constructs, namely gamification, perceived engagement, and consumer experiences. *Almarzouqi, Aburayya & Salloum (2022)* extended TAM and found that the metaverse was affected by user satisfaction, perceived trainability, user compatibility, and perceived observability. *Al-Adwan et al. (2023)* included

other external factors in the TAM, including perceived enjoyment, self-efficacy, and perceived cyber risk. However, although some studies have recently been carried out to assess the effects of certain factors on metaverse using the extended TAM, studies that incorporate other interesting factors that assist in understanding the phenomena of accepting and using metaverse are still lacking. Furthermore, it is unclear whether perceived system quality and personal innovativeness affect students' intentions toward accepting and using metaverses. Moreover, İbili et al. (2023) call for further studies to extend TAM and examine the effect of system quality and personal innovativeness on students' intention to utilise metaverse. Thus, this study proposes a theoretically extended model by extending the TAM and incorporating constructs of perceived system quality and personal innovativeness.

# RESEARCH HYPOTHESES

## TAM constructs with metaverse

Some studies have validated and confirmed the power of TAM in comprehending metaverse acceptance and use. However, empirical studies have revealed conflicting results concerning the relationship between the determinant constructs of the TAM. For instance, Almarzouqi, Aburayya & Salloum (2022) found that the PU of metaverse users was affected by PEU and that both the PU and PEU of metaverse users affected their ATT toward using metaverse (Wang & Shin, 2022). Moreover, the PU of metaverse and ATT to metaverse positively affected users' intentions toward using the metaverse (Akour et al., 2022). Additionally, perceived usefulness had a significant effect on students' intention to use metaverse (Al-Adwan et al., 2023). ATT positively affected their intentions toward using the metaverse (Aburbeian, Owda & Owda, 2022). However, Ren et al. (2022) argued that the PU of metaverse students had an insignificant effect on their intention toward using a metaverse. The argument that the ATT of metaverse did not affect students' intention to utilise metaverse has also been made. Davis (1989) demonstrated that the ATT construct had no effect on intention; thus, it was excluded from the model. Therefore, the following hypotheses were formulated:

H1: PEU of metaverse affects the PU of metaverse.

H2: PEU of metaverse affects students' intention toward using metaverse.

H3: PU of metaverse affects students' intention toward using metaverse.

## System quality (SQ)

SQ refers to qualities of speed, functions, contents, and features of metaverse. Furthermore, SQ is related to the performance of technological systems and users' evaluation of user-friendliness when using the services of these systems. It is defined as the level of assistance that users perceive in information systems (Prasetyo et al., 2021). Previous studies revealed that SQ significantly affects both PEU and PU in e-learning (Park, Nam & Cha, 2012; Rafique et al., 2020; Hawash, Mokhtar & Yusof, 2021). Additionally, the effects of SQ on user intentions to use technologies have been confirmed (Fathema & Sutton, 2013). Thus, the following hypotheses are proposed:

H4: SQ affects the PU of metaverse.

H5: SQ affects the PEU of metaverse.

## Personal innovativeness (PI)

PI stands for a user's willingness to experience a new established technology (*Agarwal & Prasad, 1998*). The term of PI is related to how users react to ideas and new approaches of doing things, with no matter of what other users have made in the past. PI has been widely utilized in literature to express and explain why users start utilizing new IT techniques or products (*Simarmata & Hia, 2020*). Some studies have confirmed the effect of PI on technologies' acceptance. For instance, *Yi, Fiedler & Park (2006)* confirmed the influence of PI on users' behavioural intentions using the following constructs: PU, PEU, and compatibility. Similarly, *Fagan, Kilmon & Pandey (2012)* found that PI significantly affected PU, PEU, and the intention to utilise virtual technology, which was designed to teach nursing students. Additionally, some previous studies have revealed that PI moderated the effect of perceived usefulness on behavior usage of IT (*Cheng, 2014*; *Shaw & Sergueeva, 2019*). Furthermore, PI had a significant effect on students' intention to use metaverse (*Al-Adwan et al., 2023*). Thus, the following hypotheses are proposed:

H6: PI affects the PU of metaverse.

H7: PI affects the PEU of metaverse.

Figure 1 illustrates the research model.

## METHODOLOGY

### Data collection

The current study assessed the factors which affect students' intentions to use metaverses. To achieve this study's objective, an extended model based on the TAM was proposed in the Saudi context. Because metaverse implementation in the educational field is in its early stages, research has enabled us to comprehend related issues by accepting and using metaverse in the field. The population of this study consist of all students enrolling in computer courses in University of Ha'il. The study participants comprised students enrolled in higher education institutes, namely University of Ha'il, who enrolled in computer courses and had knowledge of interactive technologies, such as metaverse. The positions of these participants allowed researchers to achieve the study objective. The purposive sampling was applied to select students as it targeted students who had enrolled in computer courses and had knowledge of virtual world technologies. It selects targeted participants when they fulfill specific requirements or share similar expertise with the issues under examination (*Palinkas et al., 2015*). Thus, 164 responses were collected from the University of Ha'il and validated for further analysis. According to *Boomsma (1982)*, the sample size for conducting analysis utilizing SEM should be between 100 and 200. Thus, the data obtained from the sample of 164 students were convenient and suitable for further analyses. Data were collected from 15 October 2023 to 10 November 2023 *via* questionnaires designed using Google forms and sent to participants using different online technologies, such as university emails and blackboards.

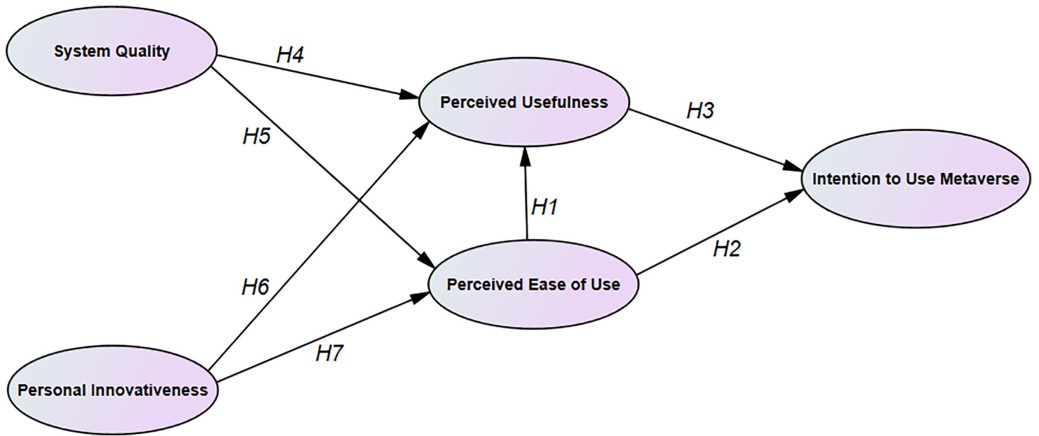

**Figure 1 The proposed research model.**

## Measures

The questionnaire comprised two main sections: the first was related to demographic information, *e.g.*, gender, academic program level, and college, while the second contained 20 items that assessed all constructs in the model. Items that measure the constructs of PU, PEU, and BI of the metaverse were adapted from a study by *Davis (1989)*. The questionnaire that measured SQ was adopted from a study by *Pham et al. (2019)*, whereas the questionnaires that measured PI were taken from a study by *Mahat, Ayub & Luan (2012)*. Word modifications and adjustments were made to the questionnaire to meet and fit the context of metaverse. While the survey language was in English, it was translated to Arabic language as participants use Arabic as main language for learning. The completed survey was first translated to Arabic, then translated back to English. The two English surveys were then compared to ensure that the translation of English to Arabic is accurate. The questionnaires included five-point Likert scales. All questionnaires are shown in Appendix 1. A written informed consent form was received from participant for this study. Moreover, This study has been reviewed and approved by the Research Ethics Committee (REC) at University of Hail dated: 13/11/2023, research no: H-2023-412.

## Data analysis

SPSS was employed to analyse the respondents' demographic information, and an SEM approach using AMOS software version 22.0 was conducted to assess the relationship between constructs and test the research hypothesis. SEM is a powerful and robust approach for assessing a proposed research model and testing research hypotheses (*Hair et al., 2012*). Furthermore, it is a well-used approach for analysing complicated models with many constructs, which makes it a better option for analysing extended theory models (*Hair et al., 2012*). Additionally, SEM is a powerful multivariate approach which is used increasingly in empirical investigation for evaluating and testing casual relationships (*Fan et al., 2016*). Following the recommendation of *Awang (2015)*, two SEM steps were applied: Confirmatory Factor Analysis (CFA) for measurement model development and SEM for testing the hypotheses.

**Table 1  Respondents' demographic information.**

|  |  | Frequency | Percent |
|---|---|---|---|
| Gender | Male | 88 | 53.7 |
|  | Female | 76 | 46.3 |
| Academic level | Bachelor's | 156 | 95.1 |
|  | Diploma | 8 | 4.9 |
| College | Business administration | 53 | 32.3 |
|  | Science | 24 | 14.6 |
|  | Computer science | 5 | 3.0 |
|  | Nursing | 11 | 6.7 |
|  | Education | 32 | 19.5 |
|  | Art | 31 | 18.9 |
|  | Applied college | 8 | 4.9 |
|  | Total | 164 | 100.0 |

## Demographic information

A total of 164 students participated in filling the questionnaires and their responses were used for further analyses. Table 1 displays the students' demographic information regarding their gender, academic level, and college. Participants included 88 males (53.7%) and 76 females (46.3%). Regarding their academic level, 156 (95.1%) students were registered in bachelor's program, while 8 (4.9%) were registered in a diploma. Regarding the colleges, 53 (32.3%) students were enrolled in the Business Administration College, 32 (19.5%) in the College of Education, 31 (18.9%) in the College of Art, and 24 (14.6%) in the College of Science, while 5 (3.0%) were enrolled in the College of Computer Science.

## RESULTS

### Confirmatory factor analysis

CFA is the best approach for validating the measurement model because of its ability to consider all types of measurement correlations and errors between constructs (*Hair et al., 2012*). Besides, it can analyse multiple constructs at the same time as treatment and assist in solving identification issues owing to the few items that measure the latent constructs. Furthermore, during CFA, three types of validity—construct, convergent, and discriminant—should be assessed (*Awang, 2015*).

Construct validity was attained when all the model fitness indices met the threshold values suggested by previous studies. Thus, CFA was conducted, and all fitness indices of the model passed the threshold values recommended by prior scholars. Figure 2 presents the CFA output.

As we can see from the results presented in Table 2, all model fitness indices have met the values suggested by previous researchers.

Therefore, convergent validity must be assessed and is attained once all values of average variance extracted (AVE) is greater than 0.5 and composite reliability (CR) is greater than 0.60 (*Awang, 2015*; *Hair et al., 2012*). Thus, convergent validity was attained

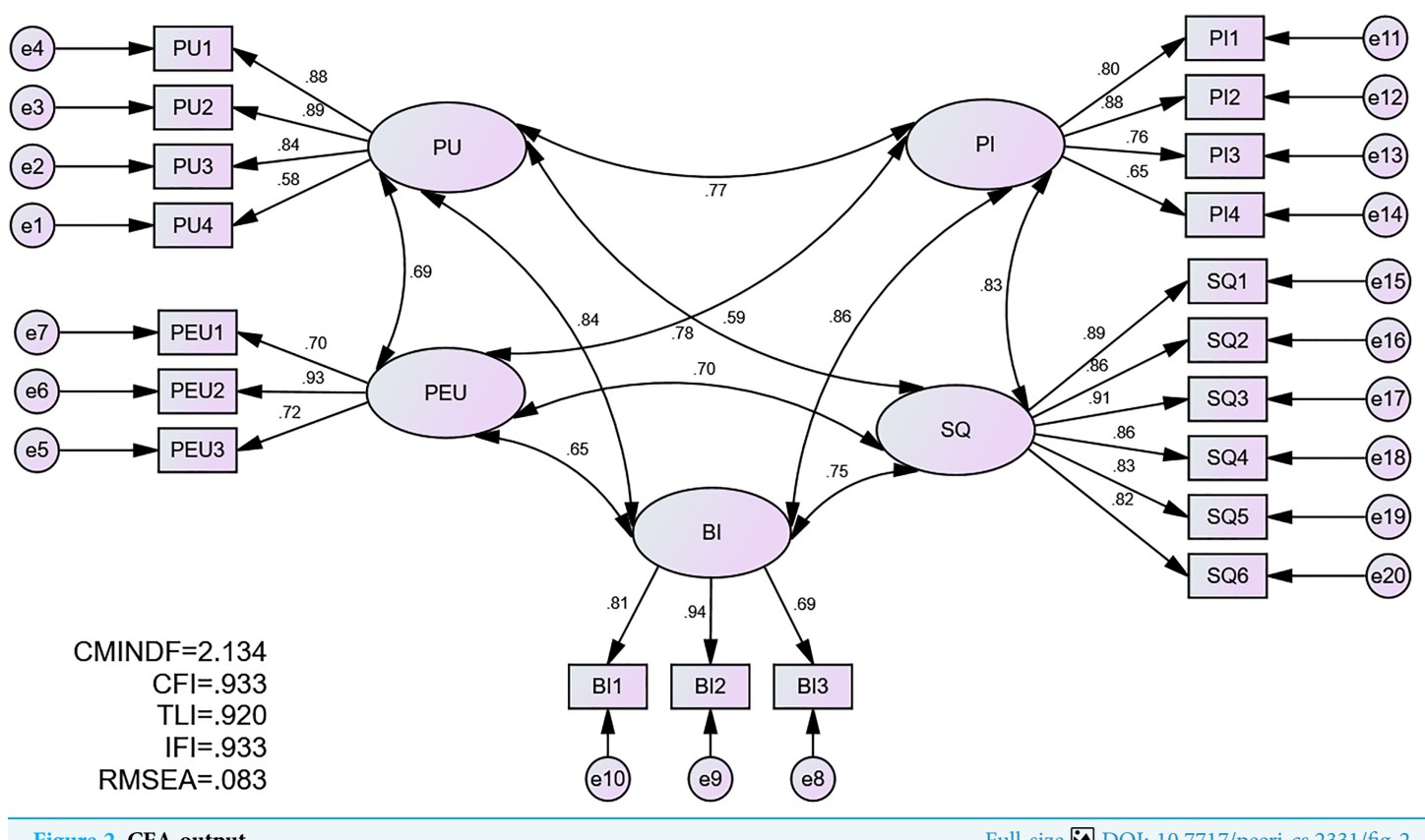

CMINDF=2.134
CFI=.933
TLI=.920
IFI=.933
RMSEA=.083

**Figure 2  CFA output.**

**Table 2  Model fitness.**

| "Name of category" | "Index name" | "Index value" | "Acceptance value" | "Decision" | "Reference" |
|---|---|---|---|---|---|
| Absolute fit | RMSEA | 0.083 | <0.1 | Accepted | *Fabrigar et al. (1999)*, *Awang (2015)* |
| Incremental fit | CFI | 0.933 | <0.90 | | |
| | TLI | 0.920 | <0.90 | | |
| | IFI | 0.933 | <0.90 | | |
| Parsimonious fit | Chisq/df | 2.134 | <3.0 | | |

as the AVE and CR values exceeded the suggested values. Table 3 presents the AVE and CR values.

Finally, discriminant validity must be assessed to ensure that all constructs of the model are discriminant and different from other constructs. In Table 4, values conveyed in bold refer to the AVE square root, and other values refer to the correlations of the constructs. Discriminant The confirmation of discriminant validity is considered when all values in bold are greater than the others in their own rows and columns (*Awang, 2015*). Therefore, it is confirmed because all values in bold are greater than the others in their own columns and rows. Table 4 presents the discriminant validity values.

**Table 3  Values of CR and AVE.**

|  | CR | AVE |
|---|---|---|
| PI | 0.858 | 0.604 |
| PU | 0.905 | 0.760 |
| PEU | 0.831 | 0.625 |
| BI | 0.858 | 0.671 |
| SQ | 0.945 | 0.743 |

**Table 4  Discriminant validity.**

|  | PI | PU | PEU | BI | SQ |
|---|---|---|---|---|---|
| PI | **0.877** |  |  |  |  |
| PU | 0.768 | **0.872** |  |  |  |
| PEU | 0.586 | 0.682 | **0.790** |  |  |
| BI | 0.857 | 0.834 | 0.653 | **0.869** |  |
| SQ | 0.831 | 0.769 | 0.698 | 0.755 | **0.862** |

**Note:**
Bold values indicate the AVE square root.

## Standardised estimates

A standardised estimate was applied to assess the beta coefficient among the latent constructs, the R value, and the loading of factors into the constructs. The standardised estimate was run first, and its output is presented in Fig. 3.

The R-squared—shown in the dependent variable of behavioural intention toward using Metaverse—is 0.76, which means that the behavioural intention to use metaverse is assessed by all related constructs in the proposed model, namely, PU, PEU, SQ, and PI. The results of R-squared show the model's high explanatory power in explaining what factors could affect students' intention toward using metaverses. *Cohen (1988)* highlighted that values of R-square higher than 0.25 confirm a high model's explanatory power. Thus, the proposed model's value of 0.76 demonstrates high explanatory power in examining the phenomenon of the factors that could contribute to students' intention toward using metaverses.

## Unstandardised estimate

An unstandardised estimate is necessary for calculating regression weight (beta estimate) and assessing critical ratio for testing the hypothesis. It was conducted and its outcome shown in Fig. 4.

The results demonstrate that SQ had a significant effect on PEU ($\beta = 0.704$, $p < 0.05$), and both PI and PEU had a significant effect on PU ($\beta = 0.505$, $p < 0.05$; $\beta = 0.235$, $p < 0.05$). Furthermore, PU had a significant effect on behaviour intention to utilise metaverse ($\beta = 0.768$, $p < 0.05$). Hence, H1, H3, H5, and H6 were supported. The results also demonstrated that PI had an insignificant effect on PEU ($\beta = 0.026$, $p > 0.05$), SQ had an insignificant effect on PU ($\beta = 0.197$, $p > 0.05$), and PEU had an insignificant effect on

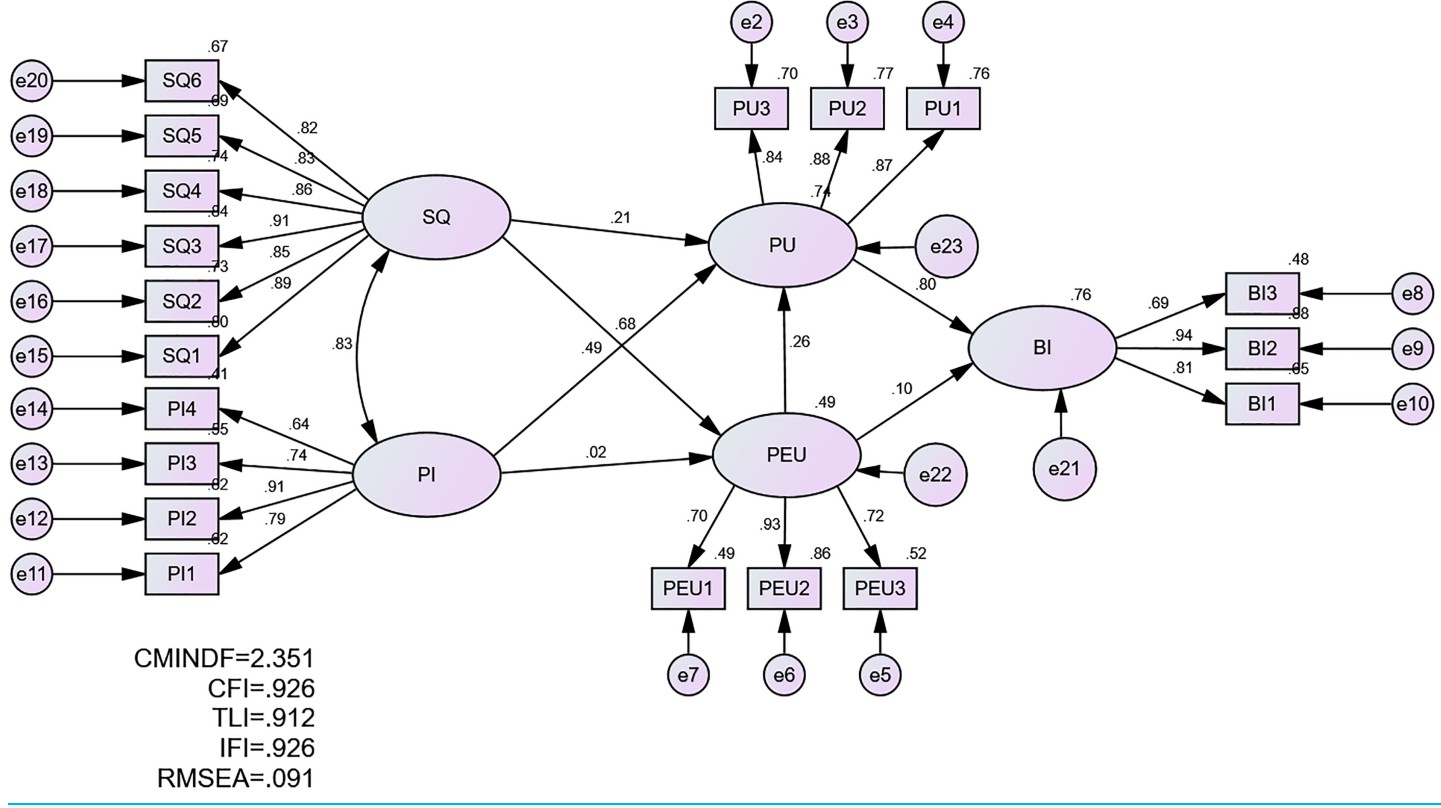

**Figure 3 Standardised estimate.**

behaviour intention to utilise metaverse ($\beta$ = 0.084, $p > 0.05$). Thus, H7, H4, and H2 are rejected. Table 5 presents the hypotheses testing results.

## DISCUSSION

The current study aimed to assess the constructs that may affect students' intentions toward using metaverses. To achieve this, the TAM was extended to incorporate two external factors: SQ and PI. By applying a structural equation modelling approach analysis, findings revealed that these two external factors were essential determinants of metaverse adoption and use. Furthermore, SQ affected students' behavioural intentions to use metaverse media through its effect on PEU. Moreover, PI affects students' intention to use metaverse technology through its effect on PU. Regarding TAM constructs, PU affected students' intentions to use the metaverse. Surprisingly, PEU did not affect effect students' intention to use metaverse media.

The findings demonstrated that SQ had a significant effect on their intention to utilise metaverses through PEU. These findings are consistent with those of previous studies which examined other technologies (*Fathema & Sutton, 2013*; *Man et al., 2020*; *Park, Nam & Cha, 2012*; *Salloum et al., 2019*). The findings indicate that all system quality aspects, such as fixability, ease of accessing information, and the available capacity in systems to

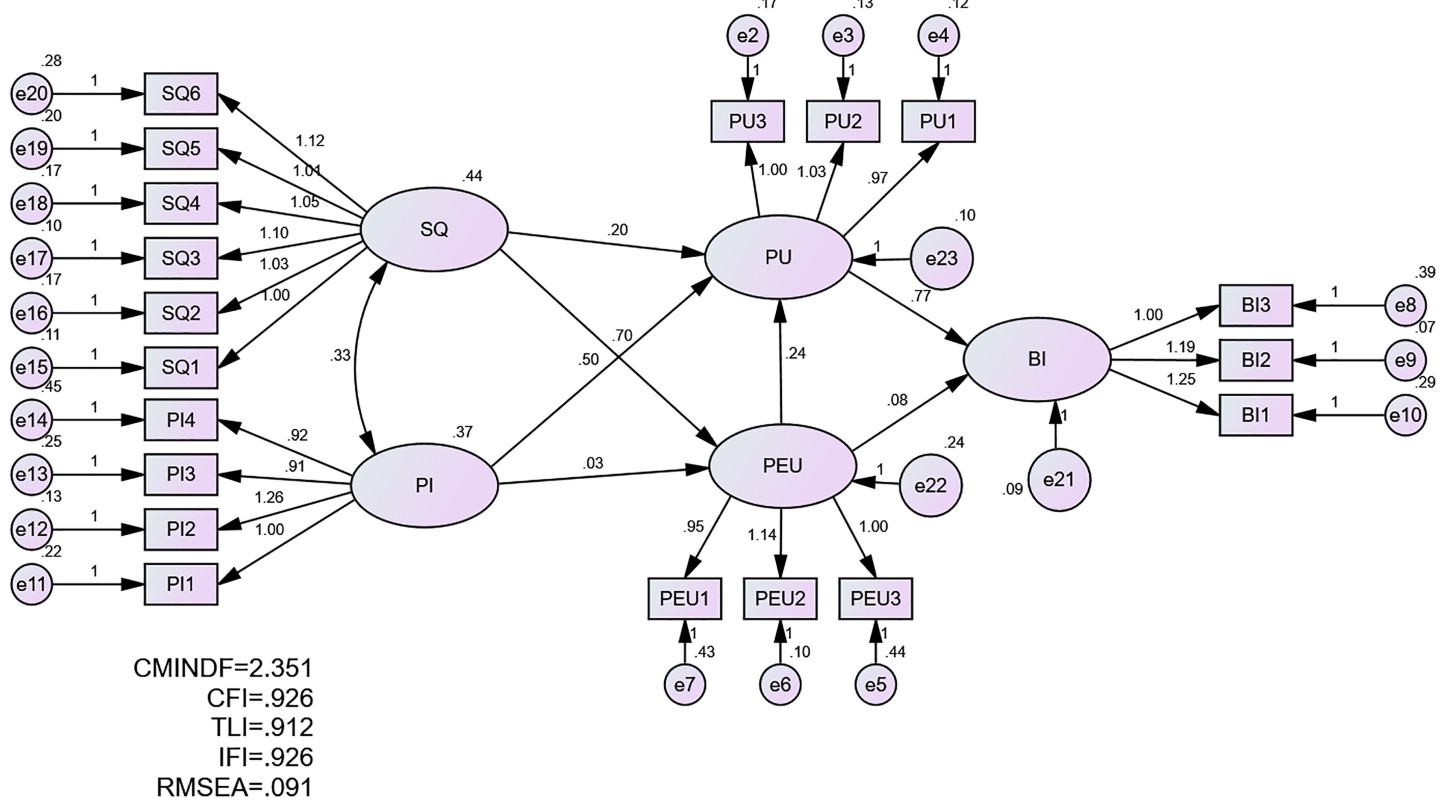

**Figure 4 Unstandardised estimate.**

**Table 5 Results of hypotheses testing.**

|  |  |  | Estimate | S.E. | C.R. | p | Results | Hypothesis | Decision |
|---|---|---|---|---|---|---|---|---|---|
| PEU | <— | SQ | 0.704 | 0.155 | 4.543 | *** | Significant | H5 | Supported |
| PEU | <— | PI | 0.026 | 0.159 | 0.166 | 0.868 | Insignificant | H7 | Rejected |
| PU | <— | SQ | 0.197 | 0.117 | 1.681 | 0.093 | Insignificant | H4 | Rejected |
| PU | <— | PI | 0.505 | 0.120 | 4.195 | *** | Significant | H6 | Supported |
| PU | <— | PEU | 0.238 | 0.076 | 3.149 | 0.002 | Significant | H1 | Supported |
| BI | <— | PEU | 0.084 | 0.071 | 1.184 | 0.236 | Insignificant | H2 | Rejected |
| BI | <— | PU | 0.768 | 0.106 | 7.212 | *** | Significant | H3 | Supported |

Note:
*** $p \leq 0.001$.

meet users' needs and expectations, are essential and contribute to perceiving a metaverse as easy for using, which then affects their intention to use it.

Furthermore, the findings showed that students' PI had a significant effect on their intention to utilise metaverses through their effect on PU. These findings are consistent with those of previous studies (*Akour et al., 2022*; *Bubou & Job, 2022*; *Al-Adwan et al., 2023*). These findings indicate that innovative students are eager to perceive a metaverse as useful, which affects their intention toward using it. This also indicates that students with

higher innovativeness tend to be more likely open to using and trying new technologies. Therefore, the higher level of students' PI affected their intention to use metaverse technology through their perception of it as useful. Moreover, students who have a higher PI level tend to perceive the metaverse as useful, which enhances their learning. Students with higher PI are often more capable of gaining benefits from using the metaverse in the early stages of implementation.

Surprisingly, the direct effect of PEU on BI was insignificant. Furthermore, PEU affects PU. These results are consistent with those of previous studies (*Wang et al., 2022*; *Yang, Ren & Gu, 2022*). However, these results contradicted the findings of other studies (*Akour et al., 2022*; *Faqih & Jaradat, 2021*). These findings were explained by the study respondents' categorisation as Generation Z (Z Gen), *i.e.*, they were born in the period from 1995 to 2010 (*Meet, Kala & Al-Adwan, 2022*). Users of this generation are considered digital natives. These students were born during the digital era and used new technologies heavily in their daily lives (*Larionova et al., 2018*). Therefore, most students had considerable expertise and knowledge in advanced modern technological applications that use AR and VR equipment. Thus, they may have only a few issues with operating and using a metaverse. In other words, perceiving a metaverse as easy to use in the digital age does not necessarily affect students' intentions toward using it. However, the findings also revealed that PEU affects PU. This indicates that students who perceive the metaverse as easy to use tend to find it useful, which assists them in gaining the benefits of metaverse during their learning.

Finally, PU had a significant effect on intentions to utilise the metaverse. The findings are consistent with those of various earlier studies (*Faqih & Jaradat, 2021*; *Zhao et al., 2022*; *Al-Adwan et al., 2023*), but inconsistent with those of others (*Yang, Ren & Gu, 2022*). These findings indicate that students are more eager to utilise the metaverse when they believe that it would be worthy and beneficial for their education and learning. Once students find using a metaverse useful, they tend to be more willing to use it for their learning.

# CONTRIBUTIONS

## Theoretical contributions

This study makes an essential contribution to the existing literature. First, it enhances our understanding of whether the acceptance and use of metaverse technology are affected by TAM constructs, among other external factors, namely, SQ and PI. Because these factors have been examined using certain technologies in some studies, their effects on the acceptance and utilization of metaverse have not been assessed. Thus, it contributes to existing literature by assessing the effects of these external factors on the acceptance and utilization of metaverse. Second, the current study contributes to TAM research by proposing a robust theoretical model that has high explanatory power (R-Square = 0.76). Furthermore, it applied a two steps in structural equation modelling approach using AMOS, which is considered a second-generation analysis technique, and confirmed the reliability and validity of the TAM in a metaverse context. CFA was conducted to validate the measurement model in terms of its construct, convergent and discriminant validities,

while SEM was applied for testing the hypothesis. Finally, this study could be utilized as a benchmark for similar future exploration in the metaverse context and offer further implications for higher education.

## Practical contributions

The study findings have important implications for practitioners and designers of metaverses. For instance, considering the effect of incorporating the constructs of SQ and PI on students' intention to use metaverses is essential. Because the SQ affected students' intention to use metaverse, through PEU, designers and practitioners should ensure improving the system quality of metaverse in regard of making the interface of the application as easy as possible, more flexible, and easier to access learning information. In addition, they should ensure the available capacity in systems to meet users' needs and expectations. Additionally, PI had a significant effect on intention to utilize metaverse through their PU. Thus, practitioners should enhance students' PI by increasing their motivation toward using and trying new technologies, especially metaverse technologies, ensuring that they affect their intention to use it more effectively. Furthermore, to increase the metaverse utilisation, practitioners and designers must provide technical support, training, and counselling to users. This continuous support can enhance students' PI toward trying and using a metaverse on a regular basis.

Moreover, the findings showed that PU affected students' intentions to use metaverses. Thus, to increase the PU of metaverse learning, it is essential to explain to students the benefits of metaverse in enhancing their learning experiences compared to traditional approaches, such as highlighting the potential of interactive and immersive learning, learning from various electronic sources, and providing opportunities to interact with others. Designers and developers of metaverse could affect how students perceive their learning by considering that using metaverse would be helpful in accomplishing their learning tasks easily and effectively.

Furthermore, students' PEU affected their PU in the metaverse. It is essential to ensure the ease and friendly user of metaverse. If a metaverse is hard to use and navigate, users might find it useful. Therefore, it is important for designers to design a platform that provides user-friendly interface with clear tutorials and instructions that explain the features of the metaverse to make it easier and more accessible. Moreover, more support and training on how to use the metaverse could effectively ease its use. This can be achieved by providing videos, tutorials, and online assistance to guide users regarding the functionalities and features of the metaverse. Additionally, it is essential to optimise metaverse applications for combination with different electronic devices. While the use of mobile devices has increased, a need remains to ensure that the metaverse is optimised for use with multiple devices, for instance tablets and smartphones. This makes it simpler for students to access information and resources using metaverses.

## Limitations and future studies

This study has a few limitations. First, its sample size was limited to students at one university, namely University of Ha'il in Saudi Arabia. Thus, caution must be exercised

when attempting to generalise these findings to other contexts. The study is geographically limited to students from one university, which might limit the generalizability of the findings. Future research could expand the demographic and geographic diversity of the sample to enhance the external validity of the study. Additionally, future studies might be carried out in different contexts for comparing the findings. Second, the current proposed model of TAM incorporates and examines the effects of two external factors—SQ and PI—on students' intention to use metaverses. However, other mediating and moderating factors, such as age, income, and personality traits, have not been examined. Thus, future studies must examine additional mediating and moderating factors. Furthermore, this study used a purely quantitative approach, whereas using a mixed method with a qualitative approach could enhance and provide a deeper comprehensive of the factors which might affect the intention toward using a metaverse.

## CONCLUSIONS

The study focused on assessing the factors affecting students' intention to use metaverse. Thus, this study proposes a theoretical research model based on TAM that incorporates two additional external factors—SQ and PI. In total, 164 students participated in the study and responded to a questionnaire. Results showed that SQ affected students' intention to utilize metaverse through PEU. Furthermore, students' PI affected their intention to utilize metaverses through their PU. Moreover, PU has a significant effect on students' intentions to use metaverse. Surprisingly, PEU did not affect their intention to utilize metaverse media. The current findings enhance our understanding of the critical roles of SQ and PI in affecting students' intentions to utilize metaverse technology. These findings could assist practitioners and designers in enhancing the adoption of metaverses among students by considering the important role of these factors which affected students' intention to utilize metaverses.

### Funding
The authors received no funding for this work.

### Competing Interests
The authors declare that they have no competing interests.

### Author Contributions
- Sultan Hammad Alshammari conceived and designed the experiments, analyzed the data, performed the computation work, prepared figures and/or tables, authored or reviewed drafts of the article, and approved the final draft.
- Mohammed Habib Alshammari performed the experiments, prepared figures and/or tables, authored or reviewed drafts of the article, and approved the final draft.

## Ethics

The following information was supplied relating to ethical approvals (*i.e.*, approving body and any reference numbers):

Research Ethics Committee (REC) at University of Ha'il, research no: H-2023-412.

## Data Availability

The raw data are available in the Supplemental File.

## Supplemental Information

Supplemental information for this article can be found online at http://dx.doi.org/10.7717/peerj-cs.2331#supplemental-information.

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
