# Peer review of "Modelling the effects of perceived system quality and personal innovativeness on the intention to use metaverse: a structural equation modelling approach"

_PeerJ Computer Science, doi:10.7717/peerj-cs.2331_

## Round 0.1 · original submission · Major Revisions

Thank you for submitting your manuscript titled to PeerJ Computer Science. After a thorough review, we require major revisions before it can be considered for publication. The literature review needs to be more critical and comprehensive to justify the novelty of your research and distinguish it from existing studies. The introduction should be expanded to articulate the research problem and contributions clearly. More details are needed on the sample population, sampling technique, and sample size. A clearer justification for using the Technology Acceptance Model (TAM) over other frameworks is required, along with a more robust operationalization of hypotheses with recent literature.

The theoretical foundation might be strengthened. The discussion can also contextualize findings within the broader literature, enhancing theoretical and practical implications with specific, actionable strategies. Please refer to the detailed comments from the reviewers for additional guidance.

We look forward to receiving your revised manuscript.

Best regards

Reviewer 1 ·

Basic reporting

This is a well-written paper that covers a timely and interesting topic. However, there are a few suggestions to improve the quality of this good work:

Introduction Section: The introduction is too short and should be extended. Focus on providing a clearer research problematization and emphasize the main contributions of your work. This will help readers understand the significance and context of your study from the outset.

Paper Structure: It is important to add a paragraph at the end of the introduction section to outline the structure of the paper. This will guide readers through the organization of your manuscript and set clear expectations.

Methodology Section: More details are required regarding the sample population, sampling technique, and the adequacy of the sample size. Providing this information will enhance the transparency and replicability of your research.

Theoretical Foundation and Hypotheses Development: Strengthen the theoretical foundation and hypotheses development by incorporating recent and related research on the metaverse. This will provide a more robust conceptual framework for your study and align it with current scholarly discourse.
Discussion Section: Utilize the incorporated studies mentioned in the theoretical foundation to strengthen the discussion section. This will help in contextualizing your findings within the broader literature and highlight the implications of your research.

By addressing these suggestions, the overall quality and impact of your paper will be significantly improved.

Experimental design

See comments above.

Validity of the findings

See comments above.

Additional comments

See comments above.

Reviewer 2 ·

Basic reporting

The structure and language of the paper are sound and proper.
The literature review needs rework.

For specific observations, Please refer to the comments section.

Experimental design

The rationale of the study needs refinement.
The methodology has some minor revisions.

For specific observations, Please refer to the comments section.

Validity of the findings

Data Analysis is sound and appropriate.
Implications (theoretical and practical) need revision.

For specific observations, Please refer to the comments section.

Additional comments

The author/s should provide some data on metaverse, GAGR/growth rate (in general and in particular to the education sector). What are the usage and trends towards metaverse in education within Saudi Arabia? Please provide these details to contextualize your study's relevance.

The rationale of the study is not justified sufficiently. Recently, several studies have examined metaverse adoption across various contexts, including education. What distinguishes your study from these existing ones? Is citing only one study adequate to justify the addition of two new constructs? Please clarify the novelty of your research.

While the literature review includes recent and relevant papers, it needs to be more critical. Simply citing previous studies is not enough. The author/s should rework the justification for incorporating additional constructs by providing a robust rationale and supporting literature to show their relevance.

Why did you choose the Technology Acceptance Model as your theoretical framework instead of other advanced IS theories like the Unified Theory of Acceptance and Use of Technology (UTAUT)? Please justify.

The author/s have mentioned, “In the TAM, PU and PEU are considered the main determinants of ATT…..”. Why was attitude (ATT) dropped in your study?

System quality is one of the constructs of the Information System Success Model (ISSM). Why is it relevant in your context? Please explain.

The operationalization of hypotheses is weak, especially for the additional constructs. Please revisit this section and include recent papers, particularly for the new constructs.

Why did you choose purposive sampling? Provide a justification for this choice.

In sub-section 4.2, “Questionnaires that measure the constructs…..”. Please replace ‘questionnaires’ with ‘items’.

The items for the ‘Personal Innovativeness’ construct were taken from a conference proceeding paper by Mahat et al. (2012). Why did you choose this source?

Did you conduct a pilot test? What were the results of the pilot test?

How did you address common method bias in your study?

The data analysis is sound and appropriate.

The implication section needs refinement. The theoretical implications are insufficient; please elaborate on the theoretical contributions of your study. For practical implications, suggest more actionable strategies. Some of the current strategies are quite generic and could be recommended without conducting this study.

Reviewer 3 ·

Basic reporting

All the comments are included in the attached file

Experimental design

All the comments are included in the attached file

Validity of the findings

All the comments are included in the attached file

Additional comments

All the comments are included in the attached file

Annotated reviews are not available for download in order to protect the identity of reviewers who chose to remain anonymous.

---

## Round 0.2 · Minor Revisions

Dear authors,
Thank you for submitting the revised version of your manuscript titled Modelling the effects of perceived system quality and personal innovativeness on the intention to use metaverse: A structural equation modelling approach." After carefully considering the reviewers' comments, I am pleased to inform you that the manuscript is nearly ready for publication. However, some minor revisions are necessary before we can proceed.
Reviewer 1 and Reviewer 3 have both expressed their satisfaction with the revisions and recommend publication. Reviewer 2 has also acknowledged significant improvements in the manuscript but has identified a few areas that need further clarification:
1. Additional data on the growth rate of the metaverse, particularly within the education sector, should be provided.
2. The justification for incorporating the additional constructs requires further strengthening, particularly in comparison to other Information Systems theories.
3. The operationalization of hypotheses for the additional constructs needs a more thorough justification.
Given the nature of these comments, we request that you make the necessary revisions to address these points. Once these minor issues are resolved, we anticipate that the manuscript will be suitable for publication.
Please submit the revised manuscript and a detailed response to the reviewers' comments at your earliest convenience.
Best regards,

Reviewer 1 ·

Basic reporting

Thank you for submitting the revised version of your paper. The paper is now ready for publication as most of the comments have been addressed.

Experimental design

None

Validity of the findings

None

Additional comments

None

Reviewer 2 ·

Basic reporting

The structure and language of the paper are sound and proper.
For specific observations, Please refer to the comments section.

Experimental design

The operationalization of hypotheses for the additional constructs requires further justification.
For specific observations, Please refer to the comments section.

Validity of the findings

Data Analysis is sound and appropriate.

Additional comments

The author/s have revised the paper based on the provided suggestions. The structure, flow, and quality of arguments have improved now. However, some areas require further attention:

1. The author/s should provide data on the GAGR/growth rate of metaverse in general and within the education sector.

2. The justification for incorporating additional constructs needs to be strengthened. While some supporting studies have been cited, the justification is not adequate.

3. Although TAM has been justified as the theoretical framework, its advantages over other advanced Information Systems theories have not been adequately explained.

4. The operationalization of hypotheses for the additional constructs requires further justification.

Reviewer 3 ·

Basic reporting

'no comment'

Experimental design

'no comment'

Validity of the findings

'no comment'

Additional comments

The authors have adequately addressed all reviewer comments. I recommend publication.

---

## Round 0.3 · accepted · Accept

I think that the paper now is suitable for publication. Congratulations.